# A Personalized Federated Learning Method Based on Clustering and Knowledge Distillation

**Jianfei Zhang ***[ID] **and Yongqiang Shi**

School of Computer Science and Technology, Changchun University of Science and Technology, Changchun 130031, China; syq@mails.cust.edu.cn
* Correspondence: jfzhang@cust.edu.cn

**Abstract:** Federated learning (FL) is a distributed machine learning paradigm under privacy preservation. However, data heterogeneity among clients leads to the shared global model obtained after training, which cannot fit the distribution of each client's dataset, and the performance of the model degrades. To address this problem, we proposed a personalized federated learning method based on clustering and knowledge distillation, called pFedCK. In this algorithm, each client has an interactive model that participates in global training and a personalized model that is only trained locally. Both of the models perform knowledge distillation with each other through the feature representation of the middle layer and the soft prediction of the model. In addition, in order to make an interaction model only obtaining the model information from the client, which has similar data distribution and avoids the interference of other heterogeneous information, the server will cluster the clients according to the similarity of the amount of parameter variation uploaded by different interaction models during every training round. By clustering clients, interaction models with similar data distributions can cooperate with each other to better fit the local dataset distribution. Thereby, the performance of personalized model can be improved by obtaining more valuable information indirectly. Finally, we conduct simulation experiments on three benchmark datasets under different data heterogeneity scenarios. Compared to the single model algorithms, the accuracy of pFedCK improved by an average of 23.4% and 23.8% over FedAvg and FedProx, respectively; compared to typical personalization algorithms, the accuracy of pFedCK improved by an average of 0.8% and 1.3%, and a maximum of 1.0% and 2.9% over FedDistill and FML.

**Keywords:** federated learning; knowledge distillation; clustering; data heterogeneity

## 1. Introduction

Nowadays, various web services such as smart cities, smart healthcare, etc., are widely used and benefit from artificial intelligence (AI) and big data. However, their training models require large-scale data support, and considering the risk of data privacy leakage, it is not possible to collect data distributed in multiple data sources together to train powerful models. Federated Learning (FL) [1], as an emerging distributed learning paradigm under privacy constraints, can reduce the risk of data privacy leakage by disclosing only the client model parameter information instead of the raw data during multi-party collaborative training. The traditional federated learning algorithm is that the server randomly selects clients to participate during each round of training and sends down global model parameters. After receiving the model, the client iteratively trains on the local dataset until the model converges, and then uploads the local model parameters to the server. Finally, the server weights and aggregates the local model parameters according to the proportion of the client's local data quantity to obtain the global model for the next round. This approach aims to train a shared global model so that this model can perform well on the data distribution of all clients. However, in reality, the data of each client is non-independently and identically distributed (Non-IID), which makes the local models obtained from training

vary widely. In this case, it is difficult to obtain a global model adapted to each client, and the phenomenon of model bias occurs, leading to model performance degradation and difficulty in convergence. This is due to the fact that each local model is updated with local data only, and local updates for individual devices have significant differences in the parameter space. Sahu, Anit Kumar et al. [2] introduced a loss term to limit the local model update and reduce the discrepancy between local and global model. Gao, Liang et al. [3] argued that restricting the optimal direction of the local model hinders its fitting to the distribution of the local dataset, and that using the local drift variable to learn the parameter gaps between the local model and the global model would give a better result. While these methods can mitigate the data heterogeneity in federated learning to a degree, a single global model is still unable to fit the data distribution of all clients simultaneously.

Therefore, many methods take a personalized federated learning approach to address the challenges caused by data heterogeneity in federated learning through designing personalized models for each client from various perspectives. Collins, Liam et al. [4] divide the local model into a base layer and a classification layer, where the base layer participates in global sharing and the classification layer is updated only locally. Deng, Yuyang et al. [5] mix local and global models and use their correlation to learn a personalized model for each client with adaptive mixing weights. Tan, Yue et al. [6] chose to transfer abstract class prototypes between the client and the server instead of model parameters, and regulate the training of local models with global class prototypes. In Zhang, Michael et al. [7], in order for each client to acquire only the knowledge that is relevant to it, other client models that perform well on the local target are chosen to be downloaded from the server. They are filtered based on their loss in the local validation set, and personalized aggregation is performed locally.

Instead, we focus on personalized federal learning using knowledge distillation. Knowledge distillation requires the migration of knowledge from the teacher model to the student model under the same dataset. However, in federated learning, the server cannot access the data information of each client. Therefore, most current approaches to combine knowledge distillation with federated learning introduce a proxy dataset and let the local model be pre-trained on the proxy dataset. Soft predictions are generated and uploaded to the server, which is weighted and aggregated and then sent down to clients. The client uses the received global soft predictions to guide the local model to train on the private data. In this way, knowledge from other client models is migrated to the local model. However, proxy datasets are usually difficult to collect, and the distillation effect of the model depends highly on how similar the proxy dataset is to the private dataset. A higher similarity means that the proxy dataset is more consistent with the data distribution of the global view, and the distillation effect is better.

Inspired by Shen, Tao et al. [8], we choose to maintain both an interaction model and a personalized model on the client, and only the interaction model is responsible for acquiring external information. And we adopt two ways to make the interactive model and the personalized model distill each other on the private dataset, based on the middle layer feature representation and the soft prediction of the model. This breaks the limitations of the proxy dataset, and allows knowledge from other client models to be migrated to the personalized model via the interaction model, which can also be continuously updated during distillation based on feedback from the personalized model. In addition, to reduce the impact of client data heterogeneity on the distillation effect, we utilize the clustering mechanism to cluster clients, so that the interaction models of clients with similar data distributions cooperate with each other to target knowledge acquisition and avoid the interference of heterogeneous information from other clients. The mechanism utilizes the similarity in the amount of parameter variation across client interaction models to measure the degree of similarity in client data distributions [9]. This allows different types of knowledge to be shared within different clusters, further improving the performance of the personalized model. Moreover, the process does not involve the private data of the

client, ensuring information security. In summary, the main contributions of this paper are as follows:

- To address the impact of data heterogeneity on model performance in federated learning, we propose a new personalized federated learning method, pFedCK, which establishes a dual-model structure on the client and combines two model mutual distillations based on the middle layer feature representation and the soft prediction of the model. It enables each client to have its own personalized model, and improves the accuracy of the model.

- Cluster partitioning of clients is realized by using the similarity of the amount of parameter variations in interaction models instead of the differences in data distribution. The interaction models of the clients with similar data distributions are enabled to learn together, thus reducing the impact of data heterogeneity on the distillation effect and further improving the accuracy of the personalized model in the Non-IID environment.

- The performance evaluation on three image datasets, MNIST, CIFAR10 and CIFAR100, shows that the method proposed in this paper has high accuracy compared to baseline algorithms.

## 2. Related Work

### 2.1. Clustered Federated Learning

Traditional federation learning aims to train a shared global model for all clients, and personalized federation learning is used to train individual models for each client. Clustered federation learning strikes a balance between them by dividing the clients into clusters according to some specific criteria and generating a common cluster model for the clients in each cluster. This enables clients with different data distributions to be converted into multiple groups with similar data distributions as much as possible, thus reducing the impact of data heterogeneity on the model performance.

Sattler, Felix et al. [9] proposed the first iterative clustering method CFL. The server clusters clients using the cosine similarity of the client gradient in each iteration. The server clusters the clients using the cosine similarity of their gradients in each iteration. Each cluster then performs federation training individually. This allows clients with similar data distributions to jointly generate a cluster model, reducing data heterogeneity in federation learning. Ghosh, Avishek et al. [10] also takes an iterative approach to clustering, but requires the number of clusters to be specified before training. In each iteration round, the server sends down all cluster models to the client. The client re-estimates the clusters to which it belongs by minimizing the loss function, and then uses the optimal cluster model for local training. This method significantly improves the model accuracy compared to CFL, but introduces a huge communication overhead, as well as aggravates the computation at the local nodes.

Some approaches try to perform one-time clustering before training and then perform federation training in each cluster. Jamali-Rad et al. [11] provide the client with an encoder to convert the data into a latent representation. The client then passes the labels of the data back to the server. The server completes one-time clustering by maximizing the separability between data labels. Liu, Bingyan et al. [12] found that neural network middle-channel sparsity can express client data distribution information, and one-time clustering of clients using sparsity values can reduce data heterogeneity among clients. Before training begins, each node first trains a sparse representation model. Then, the sparse vectors obtained from the sparse representation model are transmitted to the server. The server then performs clustering based on the similarity between the sparse vectors of each client. Also, the number of clusters needs to be specified. In contrast to iterative clustering, if these one-time clustering algorithms generate incorrect estimates at the beginning, they cannot be corrected during the training phase.

In this paper, we choose to recursively separate two groups of clients with inconsistent descent directions based on the cosine similarity of parameter variations in the client

interaction model. The method completely separates all the inconsistent clients through multiple rounds of iterations. Adaptive generation of the final number of clusters is more flexible compared to other clustering methods. Combined with knowledge distillation, it can generate a personalized model for each client. Since the personalized model can fit the local dataset distribution better, the model has a better performance.

### 2.2. Knowledge Distillation in Federated Learning

Knowledge distillation [13] is the process of training a student model by using the predictions of a complex model (teacher) as learning objectives for a simple model (student), so that its predictions are constantly close to those of the teacher models. In this manner, knowledge is transferred to the student model. The gap between the predictions of the two models is usually measured using KL scatter, also known as soft loss. In order for students to learn more about the teacher model, the logits output $h$ of the model usually requires the application of a normalization function with a temperature coefficient $T(T > 0)$. This smoothes the probability distribution corresponding to each category so that the information carried by the negative labels is relatively amplified. The soft prediction of the model is:

$$p = \frac{\exp(h/T)}{\sum_i \exp(h_i/T)} \tag{1}$$

In federated learning, soft predictions of the model can be communicated between the client and the server, instead of model parameters or gradient information. By distilling the knowledge of the global model to the client model, the generalization problem caused by data heterogeneity in federated learning can be mitigated, making the model more adaptable to data with different distributions. However, since the data distribution of each client in federated learning is different and the server cannot access the client data, a public dataset needs to be introduced. Each client first pre-trains on the public dataset to get soft predictions. The server integrates these soft predictions to get a global consensus. Then, the client models learn this consensus on the local dataset.

Li, Daliang et al. [14] proposed combining knowledge distillation with federated learning to address data heterogeneity and model heterogeneity in federated learning. Instead of model parameters, each client sends the soft predictions of its local model on the proxy dataset to the server and aggregates them into global soft predictions on the server side. It is then sent down to each client to guide its local update, migrating the knowledge of all clients to local in this way. Lin, Tao et al. [15] proposed integrated distillation for model fusion using unlabeled data or generator-generated data for user heterogeneity in federated learning. Zhu, Zhuangdi et al. [16] chose to learn a generator from the prediction rules of the client model in order to change the dependence on public datasets for knowledge distillation in federated learning. This generator is maintained by the server, and can produce feature representations predicted by the client when given a target label. The global perspective information is then provided to the client by transmitting the generator.

All of the methods mentioned aim to improve the generalization of the global model using knowledge distillation without considering the personalization of the client model. Jeong, Eunjeong et al. [17] also take a data-free knowledge distillation approach to address the data heterogeneity and model heterogeneity in federated learning. Each client stores the average logit vectors for each label and periodically uploads these local average logit vectors to a server for integration and then shares them with other clients. A generator is also used to balance the local data distribution. Cho, Yae Jee et al. [18] designed a novel PFL framework. First, they found clients with similar data distributions for clustering, and then performed co-distillation in the cluster for personalized federated learning. However, the assistance of small unlabeled public datasets was still needed.

In this paper, we design a dual-model architecture on the client and adopt two mutual distillation approaches for personalized federated learning with model middle layer feature representation and soft prediction. This approach can better understand and explain model responses to input data and without public datasets. The personalization model and the

interaction model are trained simultaneously on the local dataset to extract information from each other. Clients are also clustered to reduce the effect of data heterogeneity between clients and improve the distillation effect.

## 3. Personalized Federated Learning Method Based on Clustering and Knowledge Distillation

In this section, we present our personalized federated learning method, pFedCK. Our pFedCK employs a dual-model structure on every client and incorporates two model mutual distillations based on both of model middle-layer feature representation and model soft prediction. In addition, pFedCK uses the similarity of parameter variations in the client interaction model as an alternative for differences in data distribution between clients. Cluster partitioning of clients is implemented so that the interaction models of clients with similar data distributions are trained together. It reduced the impact of data heterogeneity on distillation effectiveness and further improved the accuracy of personalized models in Non-IID environments. Figure 1 illustrates the specific process of pFedCK. The entire process includes 5 steps.

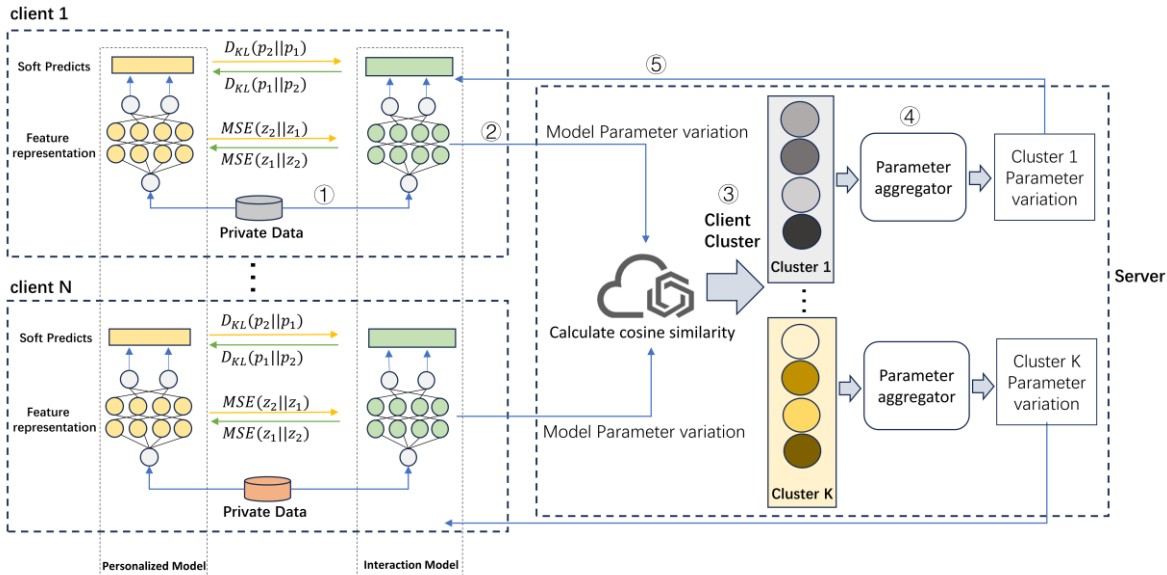

**Figure 1.** Structure of the pFedCK. Clients in different clusters have different color patterns, which means they have different data distributions. Clients in the same cluster with similar colors represent that they have similar data distributions.

Step 1: In every round of training, each client uses local data to train the personalized model and the interaction model simultaneously. It enables the interactive model to transfer information from other clients to the personalized model under the same dataset, so that the personalized model has the global perspective information. The personalized model simultaneously feeds knowledge back to the interactive model, which iteratively updates itself based on the feedback. In this way, the two models can learn from each other and progress together.

Step 2: each client calculates the parameter variations in the interaction model after training and uploads them to the server.

Step 3: the server clusters the clients according to the similarity of their parameter variations and divides the clients into different clusters.

Step 4: The server averages the parameter variations of the clients in the same cluster formed in step 3, and calculates the average value of parameter variations for each cluster.

Step 5: The server sends the average parameter variations of each cluster back to every client in the cluster. The client updates the local interaction model and start a new round of training.

In the whole process, the personalized model is only trained locally, while the interaction model participates in the global federated training process. A personalized model would be trained for every client.

### 3.1. Federated Mutual Distillation

The target of traditional federated learning methods is to cooperatively train a global model $\theta$ on a private dataset of $N$ clients such that the global objective function is minimized:

$$\min_{\theta} F(\theta) = \sum_{i=1}^{N} p_i f_i(\theta) \tag{2}$$

where $p_i$ is a predefined weight and $\sum_i p_i = 1$. The local objective function is:

$$f_i(\theta) := \mathbb{E}_{x_i \sim D_i}[L_i(\theta; x_i)] \tag{3}$$

$L_i$ is the loss function of client $i$ and $x_i$ is a randomly selected data sample from the local data distribution $D_i$. Each client receives the global model and optimizes the local objective using stochastic gradient descent on the private dataset. The updated model parameters are then uploaded to the server for weighted aggregation to obtain a new round of the global model. However, the single global model generated by this traditional method is not applicable to the case where the data of each client is heterogeneous, e.g., $D_i \neq D_j$.

The pFedCK maintains a personalized model and an interaction model at the same time. By combining the feature representations and the soft predictions of the model, the personalized model and the interaction model can distill each other during the training process. So that the interaction model transfers knowledge to the personalized model, and the personalized model can also feedback knowledge to the interaction model. Therefore, the two models can learn from each other during the training process, which effectively improves the performance of both models and eventually trains a personalized model for each client.

In pFedCK, the loss functions of the two models are as Equations (4) and (5), where $\omega_i$ denotes the interaction model of the ith client, and $\varphi_i$ denotes the personalized model trained only locally. Both of these models have the same architecture, and are initialized by clients. The interaction model $\omega_i$ shares information with other client models $\omega_j$ by participating in the global federation training, and then transmits the acquired knowledge to the personalized model $\varphi_i$ and obtains feedback from $\varphi_i$ which, in turn, continues to iterate based on the feedback.

$$L_{\varphi_i} = L_{CE_{\varphi_i}} + D_{KL}\left(p_{\omega_i} \parallel p_{\varphi_i}\right) + MSE\left(z_{\omega_i} \parallel z_{\varphi_i}\right), \tag{4}$$

$$L_{\omega_i} = L_{CE_{\omega_i}} + D_{KL}\left(p_{\varphi_i} \parallel p_{\omega_i}\right) + MSE\left(z_{\varphi_i} \parallel z_{\omega_i}\right) \tag{5}$$

where $L_{CE}$ is the cross-entropy loss function, which represents the task-specific supervision loss. $D_{KL}$ is the Kullback–Leibler (KL) scatter, which represents the knowledge used to transfer knowledge from the output soft prediction distillation loss. $MSE$ is the mean square error, which indicates the distillation loss of transferring knowledge from the middle layer feature representation of the model. $p_{\omega_i}$ and $p_{\varphi_i}$ are the soft predictions of the interaction model and the personalized model, respectively, and $z_{\omega_i}$ and $z_{\varphi_i}$ are the middle layer feature representations of the two models, respectively.

In knowledge distillation, the use of the middle layer of the model feature representation to transfer knowledge can improve the teaching effectiveness of the teacher model to the student model [19]. Because the lower layers of the model also contain important information, and this type of distillation can fully exploit the rich information in the hidden layers of the teacher model in order to encourage the student model to learn and imitate the teacher model.

In this paper, pFedCK encourages the two models to learn from each other by mutual distillation, so that their middle layer feature representations are similar to each other. It

helps the two models to be more consistent in the representation space, thus encouraging them to learn more similar features. If the mutual distillation is based only on the soft prediction of the last layer of the model, some information about the feature representation is lost. Instead, combining the two for multi-level knowledge distillation helps both parties to learn more comprehensive features and improve their understanding of the input data. In this way, the two models mimic both each other's middle layer features and each other's outputs on a training dataset with defined objectives, which will result in better performance than training each other individually. Since pFedCK maintains two structurally identical models locally and the two models are trained simultaneously, the computational overhead is double compared to traditional federated learning where one model is trained locally. Simultaneously training two models locally to distill each other does not involve communication overhead with the server. Because the process is performed independently on the local device, mutual distillation between models does not require communication in the federated learning framework. The whole learning process involves no additional communication overhead, only uploading and downloading model parameter variations. Therefore, the communication overhead between the client and the server is the same as the traditional federated learning method.

### 3.2. Clustering Based on Model Parameter Variations

In typical federated learning, a common approach to share information among clients is to upload local model parameters to the server, which then weights and aggregates the client local model parameters to produce a global model, and the server sends the global model down to all clients for local training. However, due to the heterogeneity of data across clients, the global model may perform worse than the local personalized model, and training the receiving global model from scratch for each client will also lead to a decrease in convergence speed. Meanwhile, if the server simply aggregates the parameters of the interaction model uploaded by the client to form a global model, and then the client uses the received global model to migrate knowledge to the local personalized model, it will not be possible for the personalized model to obtain more valuable information from an immature teacher model by means of knowledge distillation. Because the global model cannot fit the data distribution of all clients in the data heterogeneous environment.

Thus, in pFedCK, the client with a dual-model structure always exploits the interaction model to communicate information with other clients. And, considering the above problems, pFedCK uploads only the parameter variations of the client interaction model after training and completes the cluster division based on the similarity of the parameter variations in the interaction models of the clients. As the different data distributions of clients cause their interaction model parameters to be updated in different directions, so the similarity of parameter variations can indicate the similarity of data distributions.

pFedCK clusters the clients by measuring the cosine similarity between the parameter variations after training of each client interaction model. The interaction model only absorbs information from clients that are similar to the local data distribution, thus indirectly enabling the personalization model, which is only trained locally, to acquire more valuable knowledge from the interaction model and improve the personalization performance. The cosine similarity of clients $i$ and $j$ is:

$$S_{i,j} = \frac{\triangle \omega_i \cdot \triangle \omega_j}{\|\triangle \omega_i\| \cdot \|\triangle \omega_j\|} \tag{6}$$

where $\triangle \omega_i = \omega_i^t - \omega_i^{t-1}$ represents the parameter variations in the interaction model of client $i$ from round $t - 1$ to round $t$. The server calculates the similarity $S_{i,j,i \neq j}$ between the interaction models of each client in this way, and then constitutes the similarity matrix $S_{n \times n}$. On this basis, the division of client clusters is accomplished by using K-Means clustering. To ensure higher robustness of the algorithm, we use an iterative clustering approach, which can adaptively adjust according to the distribution and characteristics of the data, and

performs better when facing complex datasets. First, the model should be trained iteratively using traditional federated learning methods until a stable point of convergence is reached. Then, the clients are divided into clusters. Adaptive generation of the final number of clusters can adequately separate the outliers in the cluster. All these greatly ensure the accuracy of clustering. We set the number of clusters per recursive division to two, and the algorithm starts to make clustering decisions according to Equations (7) and (8) after a certain number of rounds of training.

$$\max_{i \in C} \| \triangle \omega_i \| > \varepsilon_1, \tag{7}$$

$$\operatorname*{mean}_{i \in C} \| \triangle \omega_i \| < \varepsilon_2 \tag{8}$$

If the maximum distance between client parameter variations within a cluster is greater than $\varepsilon_1$, and the average distance between client parameter variations within a cluster is less than $\varepsilon_2$, where $\varepsilon_1$ and $\varepsilon_2$ are given thresholds, the server divides the initialized list of clusters $C = (1 \cdots N)$ into $C = (c_1, c_2)$. $N$ clients will be clustered into two clusters $c_1, c_2$, respectively. In the next iteration, the clusters in the cluster list are then determined separately, and the clusters that satisfy the clustering conditions are clustered and divided to two new clusters until they do not satisfy the clustering conditions or the defined number of iteration rounds is reached. Traditional one-time clustering methods may require constant adjustment of the number of clusters to accommodate changes. This iterative clustering eliminates the need to set the final number of clusters and adaptively divides clients with similar data distributions together. After the clustering is finished, the server will average the parameter variations in the client interaction models in the clusters to aggregate them separately, so as to realize the sharing and communication of client information in the clusters. Then, the average parameter variations in each cluster are obtained,

$$\overline{\omega}_{c_k} = \frac{1}{\left| S_{c_k} \right|} \sum_{i \in c_k} \triangle \omega_i \tag{9}$$

where $c_k$ represents the $k$th cluster, and $\left| S_{c_k} \right|$ is the number of clients in the $k$th cluster. By merging the information of clients in each cluster, the interactive model can more quickly learn the underlying patterns of the overall data in the cluster, thereby achieving faster convergence during the training process. After receiving the average parameter variations in the corresponding cluster sent from the server, the client updates the interaction model parameters $\omega_i = \omega_i + \overline{\omega}_{c_k}$, $i \in c_k$ and then performs local training. In this way, the interaction model obtains more valuable information from clients in the same cluster, avoids the interference of irrelevant information, can better adapt to the local data distribution, and improves the distillation effect. Therefore, the personalized model can indirectly obtain relevant knowledge of clients of the same category by learning and imitating the middle-layer feature representation and soft prediction of the interactive model to improve its own performance.

### 3.3. The Process of pFedCK

The pseudo-code of pFedCK is shown in Algorithm 1. The whole process consists of two parts: the server side and the client side. The server is mainly responsible for receiving the parameter variations of the client interaction model and clustering the clients. In the server execution part, lines 2–4 represent the server receives the parameter variations of the client interaction models. Lines 5–10 represent the calculation of the cosine similarity between the parameter variations of all the client interaction models and the formation of the similarity matrix. Lines 11–16 represent the conditional judgment for each cluster in the cluster list. If the condition is satisfied, K-Means clustering is performed on the clients in the cluster to divide the clients that originally belonged to the same cluster into two clusters. Line 17 represents the updating of the whole cluster list after each round of clustering.

Lines 18–22 represent the average aggregation of the client interaction model parameter variations in each cluster to get the average parameter variations of the whole cluster.

The client is mainly responsible for the local training of the two models. In the client execution section, line 1 represents that each client receives the average parameter variations of the cluster where it is located and updates the interaction model parameters. Lines 2–5 represent the simultaneous training of the personalized model and interaction model on the local dataset to complete the mutual distillation operation between models. Line 6 represents the computation of parameter variations of the interaction model after training.

---

**Algorithm 1.** pFedCK

---

**Input:** $n$ clients, Set of clusters $\mathbb{C} = \{\{1, 2 \ldots n\}\}$, $L_{\omega_i}$: Loss function of interaction model, $L_{\varphi_i}$: Loss function of personalized model, initial interaction model $\omega_i$, initial personalized model $\varphi_i$, $\eta_\omega$: interaction model learning rate, $\eta_\varphi$: personalized model learning rate, number of iterations T
**Output:** $\{\varphi_i\}_{i \in n}$
**Server executes:**
1: **for** each round $t = 1, 2 \ldots T$ **do**
2:      **for** each client $i$ **in parallel do**
3:          $\triangle \omega_i^t \leftarrow$ **ClientUpdate**$(\overline{\omega}_{c_k}^{t-1})$ , $i \in c_k$
4:      **end**
5:      **for** $i = 0, 1 \ldots n$ **do**
6:          **for** j = 0, 1 \ldots n **do**
7:              $S_{i,j} \leftarrow \dfrac{\triangle \omega_i \cdot \triangle \omega_j}{\|\triangle \omega_i\| \cdot \|\triangle \omega_j\|}$
8:              $S[i][j] \leftarrow S_{i,j}$
9:          **end**
10:      **end**
11:      **for** $c \in \mathbb{C}$ **do**
12:          **if** $\max\limits_{i \in C} \|\triangle \omega_i\| > \varepsilon_1$ and $\operatorname*{mean}\limits_{i \in C} \|\triangle \omega_i\| < \varepsilon_2$
13:              $c_1, c_2 \leftarrow$ K-Means $(S_{i,j})$ , $i \in c_1, j \in c_2, c_1 \cup c_2 = c$
14:              $c = \{c_1, c_2\}$
15:          **end**
16:      **end**
17:      **Updata** $\mathbb{C}$
18:      **for** each cluster $c_k \in \mathbb{C}$ **do**
19:          **for** $i \in c_k$ **do**
20:              $\overline{\omega}_{c_k}^t \leftarrow \frac{1}{|S_{c_k}|} \sum_{i \in c_k} \triangle \omega_i^t$
21:          **end**
22:      **end**
23: **end**
**ClientUpdata:**
1: $\omega_i^t \leftarrow \omega_i^{t-1} + \overline{\omega}_{c_k}^t$ , $i \in c_k$
2: **for** each epoch $e = 1, 2 \ldots E$ **in parallel do**
3:      $\omega_i^{t+1} \leftarrow \omega_i^t - \eta_\omega \nabla L_{\omega_i}(\omega_i^t, D_i)$
4:      $\varphi_i^{t+1} \leftarrow \varphi_i^t - \eta_\varphi \nabla L_{\varphi_i}(\varphi_i^t, D_i)$
5: **end**
6: $\triangle \omega_i^{t+1} \leftarrow \omega_i^{t+1} - \omega_i^t$

---

## 4. Experiments and Analysis

In this section, we will introduce the experimental procedure and analyze the simulation results. pFedCK will be compared with two traditional federated learning algorithms, FedAvg [1] and FedProx [2], and two personalized federated learning algorithms, FML [8] and FedDistill [17], in terms of convergence and client-side accuracy. We also conduct ablation experiments on two relatively complex datasets to verify the effectiveness of different modules in pFedCK for personalized model accuracy improvement. Our development environment used Python (version 3.9) and PyTorch (version 1.12.1), and hardware-accelerated with a single NVIDIA 3060 GPU.

### 4.1. Datasets

We used three image classification datasets for our experiments, MNIST [20], CIFAR10and CIFAR100 [21]. MNIST is a dataset for handwritten digit image classification containing 60,000 training examples and 10,000 test examples with an input data size of $28 \times 28$. The CIFAR10 dataset is used to classify ten objects such as cats, birds and airplanes with 6000 examples under each category, including 5000 training examples and 1000 test examples, and the size of the input data is $32 \times 32$. The CIFAR100 dataset is an extension of the CIFAR10 dataset and contains 100 different categories, each with six hundred $32 \times 32$ pixel color images, CIFAR100 is also divided into a training set and a test set, where the training set contains 50,000 images and the test set contains 10,000 images.

### 4.2. Data Partition Setting

In this paper, we set up two different heterogeneous data scenarios. The first one is to simulate a pathological client data distribution [1]. Data samples of 2/2/10 classes are randomly sampled for each client from the MNIST/CIFAR10/CIFAR100 dataset with a total number of 10/10/100 classes, and are not duplicated with each other. The second one simulates the practical client data distribution [22]. Use Dirichlet distribution to generate non-independent identically distributed data for each client, denoted as $Dir(\alpha)$. The density function of Dirichlet is Equation (10):

$$Dir(X|\alpha) = \frac{1}{B(\alpha)} \prod_{i=1}^{K} X_i^{\alpha_i - 1} \tag{10}$$

where $X = (X_1, X_2 \ldots, X_K)$ is a K-dimensional random vector obeying the Dirichlet distribution. $B$ is the multivariate beta function defined as $B(\alpha) = \frac{\prod_{i=1}^{K} \Gamma(\alpha_i)}{\Gamma(\sum_{i=1}^{K} \alpha_i)}$. In the Dirichlet distribution, the parameter $\alpha$ regulates the probability distribution of the generated samples on each category which, in turn, affects the sampling probability of each category label in the dataset. The proportion of samples from different categories in each client's dataset is controlled to achieve the division of data and simulate the real scenario. The smaller $\alpha$ indicates the stronger data heterogeneity among clients, and the experiments consider extreme heterogeneous scenarios and set $\alpha = 0.1$. We use test data with the same distribution as the training data to compute client test accuracies, using 25% of the local data as the test dataset and the remaining 75% for training. The performance of the personalized federated learning algorithm is evaluated by the average test accuracy of the local model across all clients on the local test set. The performance of the traditional federated learning algorithm is evaluated by the average test accuracy of the global model across all clients on the local test set.

### 4.3. Parameter Settings

If not specifically stated, the experiments will use the following hyperparameter settings. We use a four-layer CNN network [1] for the image classification task, which contains two $5 \times 5$ convolutional layers. The first convolutional layer has 32 channels, followed by a ReLU activation function and a $2 \times 2$ maximum pooling layer. The second convolutional layer has 64 channels, also followed by a ReLU activation function and a $2 \times 2$ maximum pooling layer. This is followed by a fully connected layer with 512 units and ReLU activation. Finally, the final classification result is output through a linear layer. The optimizer for both personalized and interactive model training uses the SGD algorithm, with a learning rate of 0.01 for the personalized model, a learning rate decay coefficient of 0.99, and a learning rate of 0.005 for the interactive model, with the data batch size set to 32, and the local training epochs is 5, for a total of 100 rounds of training. The number of clients is 20, and the client participation rate $\rho = 1$. For the MNIST and CIFAR10 datasets, the maximum distance threshold between client parameter changes within clusters is set $\varepsilon_1 = 0.3$, and the average distance threshold between client parameter changes within clusters is set $\varepsilon_2 = 0.04$, and for CIFAR100, $\varepsilon_1 = 0.3$ and $\varepsilon_2 = 0.08$.

*4.4. Baseline Algorithm*

FedAvg: a traditional federated learning algorithm where the client and server send model parameters to each other, and the global model is aggregated on the server using a weighted average.

FedProx: using global model parameters as regularization terms on the basis of FedAvg, and restricting the direction of local model updating so that the aggregated global model better fits the local dataset distribution.

FML: a personalized federated distillation algorithm that does not depend on the proxy dataset and participates in federated learning with shared model parameters.

FedDistill: data-free federated distillation algorithms that do not share model parameters, but instead share label averages of logit vectors with other clients.

*4.5. Results and Discussion*

We evaluated the performance of all algorithms in two extremely heterogeneous scenarios. Table 1 demonstrates the accuracy of each algorithm on different datasets. As can be seen from Table 1, the pFedCK algorithm improves the personalization accuracy by about 2% on top of the FedAvg and FedProx, and by 0.42% and 0.07% on top of the FML and FedDistill, respectively, under the MNIST dataset in the practical heterogeneous environment. The improvement is not significant because MNIST is a simple handwritten digit dataset and the algorithms perform the classification task well even in a heterogeneous environment. However, under the CIFAR10 dataset, pFedCK improves the personalization accuracy by about 35% over the FedAvg and FedProx and by about 1.4% over the FedDistill and FML. The improvement is even more obvious with the more complex CIFAR100 dataset. pFedCK can improve 1% and 2.4% over FedDistill and FML, respectively, and also improves about 22% compared to traditional federated learning algorithms.

**Table 1.** Accuracy (%) of client personalized model testing in pathologic heterogeneity setting and practical heterogeneity setting.

|  | Practical Heterogeneous | | | Pathological Heterogeneous | | |
|---|---|---|---|---|---|---|
| **Methods** | **MNIST** | **CIFAR10** | **CIFAR100** | **MNIST** | **CIFAR10** | **CIFAR100** |
| FedAvg | 97.36 | 53.06 | 27.36 | 93.35 | 53.38 | 22.69 |
| FedProx | 97.27 | 52.37 | 26.92 | 92.71 | 52.66 | 22.42 |
| FedDistill | 99.36 | 85.73 | 47.66 | 99.78 | 87.81 | 62.68 |
| FML | 99.01 | 85.96 | 46.25 | 99.67 | 88.12 | 60.65 |
| pFedCK | 99.43 | 87.13 | 48.66 | 99.81 | 89.09 | 63.55 |

It shows that the pFedCK algorithm is more adapted to the complex data heterogeneous environment. The FedAvg and FedProx algorithms do not take into account the individualized needs of different clients, and thus the performance degrades sharply in complex data heterogeneous environments. The FedDistill algorithm's performance degradation is a non-negligible factor since it does not share network parameters. The FML algorithm does not take advantage of the richness of information in the hidden layers of the teacher model, whereas the pFedCK algorithm implements personalized federated learning by using multi-level distillation of knowledge and takes into account the effect of mutual collaboration between the clients, clustering the clients based on the amount of model parameter variations as a means to improve the personalization performance.

In pathologically heterogeneous environments, pFedCK performs better, improving by 2.9% and 0.87% over FML and FedDistill under the CIFAR100 dataset, and by about 40% over traditional federated learning algorithms. The reason is that in pathologically heterogeneous environments, each client has only a few categories of data samples. pFedCK clusters better in such environments, and can accurately divide clients with the same category of data samples together to achieve information exchange within the clusters, improve the model distillation effect, and transfer information about clients with the

same data distribution to the personalized model. In summary, in both heterogeneous environments, pFedCK shows significant improvements compared to both FedAvg and FedProx, and small improvements compared to FML and FedDistill.

Figure 2 shows the accuracy curves of pFedCK and other baseline algorithms on the MNIST dataset in the two heterogeneous environments, respectively. As can be seen from Figure 2a, the two traditional federated learning algorithms, FedAvg and FedProx, both increase in accuracy with the increase in the number of iterations. And all three personalized federated learning algorithms reach high test accuracy in the first few rounds of training, and gradually converge near Rounds = 20, with the convergence speed and test accuracy significantly higher than the two traditional federated learning algorithms. This phenomenon is more obvious in complex image processing tasks, such as the CIFAR10 and CIFAR100 datasets. Figure 2b illustrates the accuracy curves of the algorithms in a pathologically heterogeneous environment. The FedDistill and FML algorithms are basically on the same level as the pFedCK algorithm in terms of accuracy and speed of convergence. Two traditional federated learning algorithms, FedAvg and FedProx, perform even worse in pathologically heterogeneous environments, with accuracy decreasing by about 4% compared to practical heterogeneous environments, and non-convergence at the end of training. The reason is that the single global model is poorly generalized and cannot be adapted to different client data distributions. Although FedProx adds a regularization term on the basis of FedAvg to control the variation in model parameters and prevent the phenomenon of model drift, it is not ideal from the results. Compared with traditional federation learning algorithms, personalized federation learning algorithms are more suitable for data heterogeneous environments. FedDistill, FML and pFedCK are all personalized federated distillation algorithms that do not rely on external proxy datasets, and they all showed good performance. This is because they allow each client to update the model according to the characteristics of its local data. This personalized training allows each client to focus more on the characteristics of its own data, thus capturing data heterogeneity more effectively.

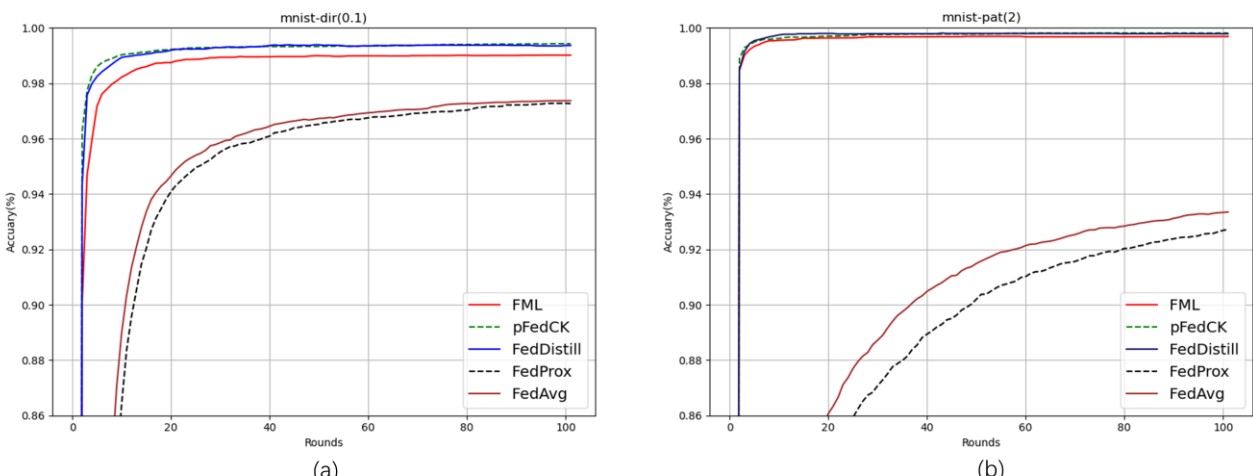

**Figure 2.** The accuracy curves for pFedCK compared to other baseline algorithms on the MNIST dataset. (**a**) Practical heterogeneous environment, (**b**) pathologic heterogeneous environment.

Since the test accuracy gap between personalized federated learning algorithms and traditional federated learning algorithms is too dramatic, we only show the average accuracy curves of three personalized federated learning algorithms on CIFAR10 and CIFAR100 datasets on Figure 3. As can be seen from Figure 3, pFedCK outperforms all benchmark algorithms in most of the scenarios. Figure 3a shows the accuracy curves of each algorithm on the CIFAR10 dataset in a practical heterogeneous environment. The pFedCK algorithm improves the personalization accuracy over the FedDistill algorithm and the FML algorithm by 1.4% and 1.17%, respectively, and the accuracy curves are flatter compared to that of

FedDistill. The FedDistill algorithm has a more obvious fluctuation during the process, and the accuracy shows a slightly decreasing trend in the end of the global iteration, which is relatively unstable.

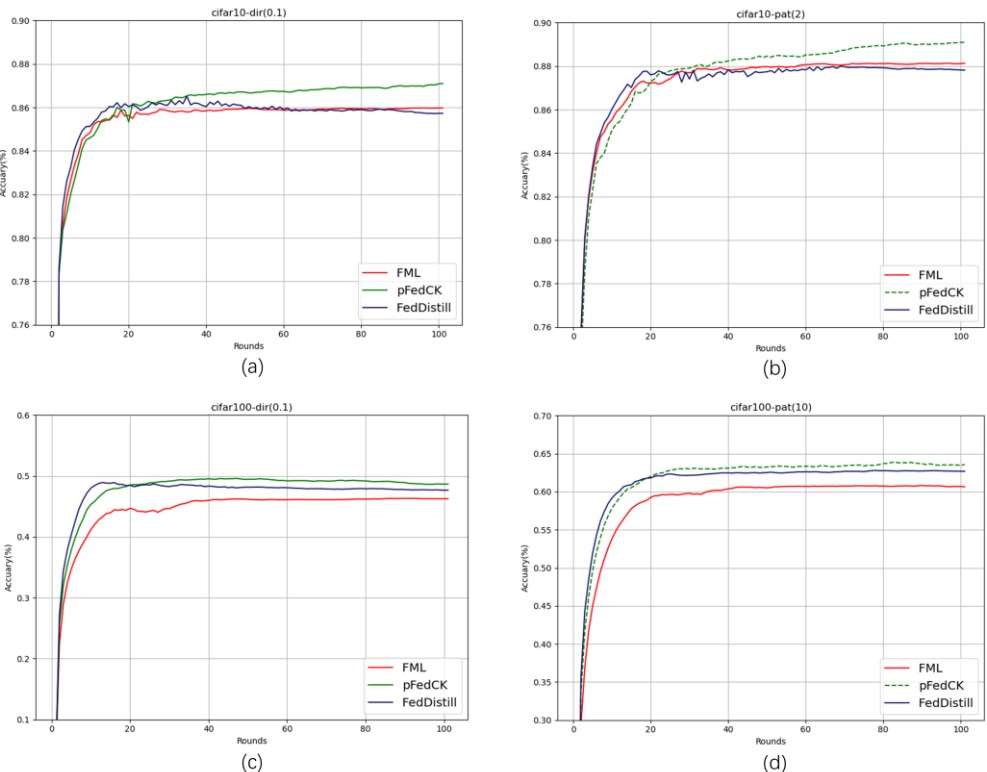

**Figure 3.** Average accuracy curves of the three personalized federated learning algorithms on CIFAR10 and CIFAR100 datasets under different heterogeneous scenarios. (**a**) Average accuracy curve of each algorithm on CIFAR10 under practical heterogeneous scenarios. (**b**) Average accuracy curve of each algorithm on CIFAR10 under the pathologic heterogeneity scenario. (**c**) Average accuracy curve of each algorithm on CIFAR100 under the practical heterogeneity scenario. (**d**) Average accuracy curve of each algorithm on CIFAR100 under the pathological heterogeneity scenario.

Figure 3b illustrates the accuracy curves of the algorithms on the CIFAR10 dataset under pathologically heterogeneous environments. It can be seen that pFedCK has slightly inferior accuracy and convergence speed compared to the baseline in the first few rounds of testing, although in different heterogeneous scenarios on the CIFAR10 dataset. However, after 20 rounds, FedDistill and FML are gradually stabilized, while the accuracy of pFedCK is still further improved. This is because pFedCK uses a combination of model middle layer feature representation and model soft prediction to allow local models and interactive models to mutually distill and learn more comprehensive and extensive feature information. And clustering begins around the 20th round, so that client interaction models with similar data distributions are trained cooperatively, which reduces client data heterogeneity and improves the distillation effect of interaction models and local models. The effect is more obvious on the more complex dataset CIFAR100.

As can be seen from Figure 3c, pFedCK is significantly better than FML in terms of convergence speed and accuracy improvement. pFedCK accuracy is also significantly better than FedDistill after the 20th round in the practical heterogeneous scenario. In the pathological heterogeneous scenario shown in Figure 3d, the training effect of pFedCK is significantly better than FedDistill in the middle and late stages of training, and the training stability is also better. In summary, under the same experimental setup, the pFedCK algorithm can obtain better results than other algorithms in different datasets and different data heterogeneous scenarios.

*4.6. Ablation Experiment*

In this section, we evaluate the effectiveness of each component in pFedCK. The version of pFedCK that does not deploy mutual distillation based on the feature representation of the middle layer of the model is represented by pFedCK-f, and the version of pFedCK that does not deploy clustering based on the interaction model variation is represented by pFedCK-c. These two versions were experimentally compared with complete pFedCK on CIFAR10 and CIFAR100 datasets in pathologically heterogeneous settings. Table 2 lists the experimental results.

**Table 2.** Accuracy of pFedCK vs. other versions on CIFAR10 and CIFAR100.

| Dataset | pFedCK-f | pFedCK-c | pFedCK |
|---------|----------|----------|--------|
| CIFAR10 | 88.95 | 88.89 | 89.09 |
| CIFAR100 | 61.77 | 63.23 | 63.55 |

As can be seen from the experimental results in Table 2, clustering is more effective on the CIFAR10 dataset, and mutual distillation based on the model middle layer feature representation is more effective on the CIFAR100 dataset. However, pFedCK-f and pFedCK-c do not perform as well as the complete version of the pFedCK algorithm on both datasets. This suggests that both the strategies of feature representation-based mutual distillation and interactive model variation clustering promote the learning effect of models in heterogeneous scenarios, which is helpful in solving the problem of data heterogeneity across clients in federated learning.

**5. Conclusions**

In this paper, we propose a personalized federated learning method based on mutual distillation and clustering, pFedCK. The pFedCK exploited a dual-model on each client, and realized mutual distillation based on soft prediction of models and intermediate feature representation of two models, prompting them to learn more similar features. It improves the generalization ability of the models and learn more abstract feature representations. On this basis, pFedCK divides clients with similar data distribution into the same cluster. Client interaction models in the same cluster can train together and effectively obtain more valuable information from others, while avoiding the interference of irrelevant information. Mutual distillation and clustering enable the interaction model to better adapt to the local data distribution, and achieve more significant results in the distillation process. In the same way, the personalized model can interact with other clients of the same category to indirectly acquire relevant domain knowledge and improve its performance.

The simulation results showed that the pFedCK significantly improves the accuracy compared to all baseline algorithm. Especially on more complex practical heterogeneous scenarios, such as the CIFAR100, pFedCK improved 21.30%, 21.74%, 2.41% and 1.00% higher than the FedAvg, FedProx, FML and FedDistill methods, respectively, and the convergence speed is also significantly improved. Since it is difficult to use the same model structure for the personalized models of each client in real scenarios, in the future, we will study the performance of pFedCK in model heterogeneous situations, such as the use of models of different sizes and structures between clients. Meanwhile, in order to further improve the robustness of clustering, we will consider the impact of client soft clustering on model performance in different heterogeneous scenarios.

**Author Contributions:** Conceptualization, J.Z. and Y.S.; methodology, J.Z. and Y.S.; software, Y.S.; validation, J.Z. and Y.S.; formal analysis, J.Z. and Y.S.; investigation, J.Z. and Y.S.; resources, J.Z. and Y.S.; data curation, J.Z. and Y.S.; writing—original draft preparation, J.Z. and Y.S.; writing—review and editing, J.Z. and Y.S.; visualization, J.Z. and Y.S.; supervision, J.Z.; project administration, J.Z. and Y.S.; funding acquisition, J.Z. All authors have read and agreed to the published version of the manuscript.

**Funding:** This paper supported by the project "Research on Machine Learning Methods Based on Multi-party Participation" (20210101483JC), which is financially supported by the Science & Technology Development Program of Jilin Province, China.

**Data Availability Statement:** The datasets that support the results of this study are publicly available datasets, and the use of these datasets in this work adheres to the licenses of these datasets. The MNIST dataset is available at http://yann.lecun.com/exdb/mnist/ (accessed on 15 August 2023). The CIFAR10 dataset is available at http://www.cs.toronto.edu/~kriz/cifar-10-python.tar.gz (accessed on 15 August 2023). The CIFAR100 dataset is now available at http://www.cs.toronto.edu/~kriz/cifar-100-python.tar.gz (accessed on 15 August 2023).

**Conflicts of Interest:** The authors declare no conflicts of interest.

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
