# Peer review of "A Personalized Federated Learning Method Based on Clustering and Knowledge Distillation"

_electronics, doi:10.3390/electronics13050857_

Round 1
Reviewer 1 Report
Comments and Suggestions for Authors
This paper comprehensively presents an interesting and promising method for personalized (clustered) federated learning for applications where this kind of approach makes sense.
Apart from some considerations regarding the quality of the language in some parts of the manuscript, there are some minor issues for the authors to address in order to improve the quality of the presentation:
1. In Figure 1, the same counting notation is used (1..N) for both clients and clusters, although their numbers are, by definition, different
2. Section 4.3 presents a specific and detailed NN configuration without any argumentation regarding the hyperparameter choices. I believe it would be helpful for these choices to be explained (in contrast with other possible NN configurations)
3. In section 4.5, given that accuracy is in %, the reporting of the improvements is a little ambiguous as it is not in every case clear if it refers to a relative improvement over the baseline value or an absolute accuracy percentage increase. Maybe reporting old and new accuracy values in place (in parentheses after the improvements are mentioned) could facilitate understanding the respective figures.
Comments on the Quality of English LanguageA careful proofreading of the final manuscript is warranted to address some minor grammatical and syntactic errors and improve the quality and flow of the text.
Reviewer 2 Report
Comments and Suggestions for Authors
This paper proposes a novel method named pFedCK for personalized federated learning, which utilizes clustering and knowledge distillation to address data heterogeneity across clients. The main novelties include the integration of a clustering mechanism for grouping similar clients and applying knowledge distillation to enhance personalized model training, thereby improving learning efficiency and performance in heterogeneous data environments.
The related work section is well designed but the authors could consider the following suggestions:
- Expand the Discussion on Clustering Techniques: In the Related Work section, the authors are encouraged to provide a more comprehensive overview of existing clustering methods used in federated learning. This should include a comparison of various approaches, their applicability in different federated learning scenarios, and how they have been utilized to address data heterogeneity. This expanded discussion will not only contextualize the authors' choice of clustering in pFedCK but also demonstrate their method's relevance and advancement over existing tecniques.
- Incorporate Recent Developments in Knowledge Distillation: The authors should consider updating the Related Work section to include recent developments and cutting-edge research in the field of knowledge distillation, particularly those relevant to federated learning. By highlighting these advancements, the authors can more effectively position pFedCK's approach to knowledge distillation and underscore its innovative aspects. This will also help in demonstrating the significance of their contributions to the field and how pFedCK advances the current state of knowledge distillation in federated learning environments.
In section 3:
- A concern with the proposed pFedCK method is the robustness of the clustering mechanism, especially in scenarios where clients have significantly diverse data distributions. There is a risk that the clustering algorithm might incorrectly group clients, leading to suboptimal training and potentially degrading the performance of the personalized models. Further exploration on how pFedCK handles misaligned clusters and its impact on model accuracy would be beneficial.
- While knowledge distillation is a key component of pFedCK, there are concerns regarding its efficiency and scalability, particularly when dealing with a large number of clients or highly complex models. The computational overhead and communication costs associated with the distillation process in a federated learning setting need to be thoroughly assessed. Addressing these concerns would strengthen the practical applicability of pFedCK in diverse federated learning environments.
in section 4:
-The Experiments and Analysis section could benefit from a more detailed description of the experimental setup, including specifics on the computational resources used, hyperparameters, and the exact process of data distributon among clients. This level of detail is crucial for ensuring the reproducibility of the results and for allowing other researchers to accurately compare pFedCK with other methods under similar conditions.
in section 5:
- The Conclusions section could be enhanced by providing a more detailed discussion on future work and potential extensions of the pFedCK method. While the section summarizes the findings, it lacks depth in exploring how the research could evolve, particularly in addressing the identified limitations or in adapting the metod to different federated learning contexts and emerging challenges.
Round 2
Reviewer 2 Report
Comments and Suggestions for Authors
The authors have satisfactorily addressed all of my concerns, and from my perspective, the manuscript has significantly improved in both quality and clarity. Consequently, I believe it is now ready for publication. This improvement reflects a thorough revision process that has enhanced the paper's coherence, making its findings and contributions to the field more accessible and compelling to the readers.